# Synthesis and Characterization of Titanium Nitride–Carbon Composites and Their Use in Lithium-Ion Batteries

**DOI:** 10.3390/nano14070624

**Published:** 2024-04-02

**Authors:** Helia Magali Morales, Horacio Vieyra, David A. Sanchez, Elizabeth M. Fletes, Michael Odlyzko, Timothy P. Lodge, Victoria Padilla-Gainza, Mataz Alcoutlabi, Jason G. Parsons

**Affiliations:** 1School of Integrative Biological and Chemical Sciences, University of Texas Rio Grande Valley, 1 West University Blvd., Brownsville, TX 78521, USA; helia.morales@utrgv.edu; 2School of Engineering and Sciences, Tecnologico de Monterrey, Av. E. Garza Sada 2501, Monterrey 64849, NL, Mexico; h.vieyra@tec.mx; 3Department of Mechanical Engineering, University of Texas Rio Grande Valley, 1201 West University Dr., Edinburg, TX 78539, USA; david.sanchez11@utrgv.edu (D.A.S.); elizabeth.fletes01@utrgv.edu (E.M.F.); mataz.alcoutlabi@utrgv.edu (M.A.); 4Characterization Facility, College of Science and Engineering, 55 Shepherd Laboratories, University of Minnesota, Minneapolis, MN 55455, USA; odlyz003@umn.edu; 5Department of Chemical Engineering and Materials Science, University of Minnesota, Minneapolis, MN 55455, USA; lodge@umn.edu; 6Department of Chemistry, University of Minnesota, Minneapolis, MN 55455, USA; 7School of Earth Environmental and Marine Sciences, University of Texas Rio Grande Valley, 1 West University Blvd., Brownsville, TX 78521, USA

**Keywords:** thermal decomposition, titanyl phthalocyanine, LIBs, TiN, TiN–carbon composite

## Abstract

This work focuses on the synthesis of titanium nitride–carbon (TiN–carbon) composites by the thermal decomposition of a titanyl phthalocyanine (TiN(TD)) precursor into TiN. The synthesis of TiN was also performed using the sol-gel method (TiN(SG)) of an alkoxide/urea. The structure and morphology of the TiN–carbon and its precursors were characterized by XRD, FTIR, SEM, TEM, EDS, and XPS. The FTIR results confirmed the presence of the titanium phthalocyanine (TiOPC) complex, while the XRD data corroborated the decomposition of TiOPC into TiN. The resultant TiN exhibited a cubic structure with the FM3-M lattice, aligning with the crystal system of the synthesized TiN via the alkoxide route. The XPS results indicated that the particles synthesized from the thermal decomposition of TiOPC resulted in the formation of TiN–carbon composites. The TiN particles were present as clusters of small spherical particles within the carbon matrix, displaying a porous sponge-like morphology. The proposed thermal decomposition method resulted in the formation of metal nitride composites with high carbon content, which were used as anodes for Li-ion half cells. The TiN–carbon composite anode showed a good specific capacity after 100 cycles at a current density of 100 mAg^−1^.

## 1. Introduction

The development of new renewable energy sources has been driven by the rapidly growing market for mobile electronic devices as well as hybrid and electric vehicles, which require highly efficient, low-cost, and environmentally friendly energy storage technologies [1,2]. Lithium-ion batteries (LIBs) have been the dominant power source for both portable electronics and electric vehicles [3]. These technologies are promising for large-scale energy storage because of their superior performance, high energy density, high specific power, long cycle life, small memory effects, low self-discharge rate, environmental friendliness, and low mass density [4,5].

Extensive research has contributed to enhancing LIB performance, affordability, and safety. These enhancements in LIB technology have been achieved largely through the improvement of the electrochemical performance of the electrode and separator materials [4,6]. In fact, the electrochemical performance of LIBs depends significantly on the properties and morphology of the electrode materials; thus, the exploration of advanced anode materials has been highly active. In addition to the more traditional intercalation-type materials, diverse alloying and conversion materials offer remarkable potential for use in LIB applications due to their high specific capacities, higher electronic conductivity, and great chemical and thermal stability [4]. Currently, graphite is the most used anode material in commercial LIBs due to its low cost, high abundance, and high cyclability. However, the graphite anode exhibits low theoretical specific capacity (372 mAhg^−1^) [4]. Therefore, it is crucial to replace the graphite anode.

Recent research has focused on alloying-type anode materials, such as silicon, tin, aluminum, magnesium, silver, iron, and antimony; furthermore, conversion-type materials, such as transition metal dichalcogenides (TMDs), transition metal oxides (TMOs), and transition metal nitrides (TMNs) have also been areas of focus. These new materials have been developed as high-performance anode materials due to their high specific capacities, as well as their chemical and thermal stability [4]. Different strategies, such as morphology control, have been used to improve the electrochemical performance of anode materials. For example, the use of nanowires, nanotubes, core-shell, coral-like, hollow, microspheres, and carbonaceous-based composites have been investigated [2,4,7,8].

The early use of nitrides in LIBs, either as lithium transition metal nitrides or binary transition metal nitrides, showed poor electrochemical performance. However, metal nitrides have attracted considerable attention due to possessing low polarization, low polarization loss, and higher electronic conductivity than transition metal oxides and the formation of Li_3_N [9]. Lithium is stored in both the crystalline and amorphous structures of TMNs through a conversion reaction [10]. The products of the conversion reaction for TMOs and TMDs are Li_2_O and Li_2_S, which are poor conductors; however, the TMNs form Li_3_N as a conversion product, which is a superionic conducting material [4,9]. The most used TMN anode materials in LIBs include CoN, TiN, VN, and CrN; other examined TMNs, such as Zn_3_N_2_, Cu_3_N, and Ge_3_N_4,_ have exhibited moderate to poor electrochemical stability [9].

TiN is an attractive material with extensive applications in coatings, catalysis, sensors for electroanalysis, and supercapacitors. Tang et al. coated silicon nanoparticles with TiN to improve the cycling performance of LIBs; the resultant anode material exhibited a reversible specific capacity of 1900 and 400 mAhg^−1^ after 100 cycles at 0.1 C and 2 C charging, respectively [11]. Similarly, Balogun et al. grew TiO_2_ nanowires on carbon cloth by a hydrothermal method and subsequent annealing at 800 °C under a nitrogen atmosphere [12]. The TiO_2_ was coated with a TiN shell, changing the flow to ammonia and keeping the temperature constant at 800 °C. The TiO_2_/TiN/C composite anode delivered a capacity of 203 mAhg^−1^ after 650 cycles at 10 C. Zheng et al. fabricated titanium nitride nanowires that supported silicon nanorods via radio frequency magnetron sputtering of Si onto the TiN nanowires [13]. After 200 cycles, the composite anode material exhibited a specific capacity of 3258 and 2256 mAhg^−1^ at a current density of 1 and 10 Ag^−1^, respectively, indicating that the TiN/Si composite can be considered a promising anode material for LIBs. Dong et al. synthesized TiN mesoporous spheres with cyanamide to prevent nanopore collapse during conversion and recrystallization [14]. A mixture of TiO_2_ spheres and cyanamide was heated to 800 °C under an ammonia atmosphere for 1 h. The energy density of the composite anode was 45.0 WhKg^−1^ at a power density of 150 Wkg^−1^, showing a potential application as a high-energy capacitor and LIB.

TiN can be synthesized by different methods such as combustion, sol-gel, magnetron sputtering, mechano-synthesis, metathesis, nitridation, ammonolysis, laser nitridation, carbothermal reduction–nitridation, electrochemically, chemical vapor deposition, physical vapor deposition, and thermal decomposition [15,16,17,18,19,20,21,22,23,24,25,26,27]. Several of these methods, which include high temperatures and long synthesis times, are either challenging or expensive [28]. However, metal–organic compounds have been valuable sacrificial templates to prepare metal-based nanoparticles dispersed in a carbon matrix [29]. The selection of the starting materials allows control over the resultant structure, generating materials with high porosity and high surface area, as well as improved electrochemical properties. Metal-substituted phthalocyanines are well-known conjugated macrocycles that form two-dimensional layered structures. This class of compounds has been shown to produce metal nitride/amorphous carbon composites [30].

The main goal of the present work is the synthesis and characterization of TiN(TD) by the thermal decomposition of titanyl phthalocyanine precursor as well as the possibility of using TiN/C composite as an anode for LIBs. The current manuscript discusses in detail the synthesis and characterization of TiN as well as the synthesis of carbon from the H_2_PC. Further work will be conducted to investigate the electrochemical performance of these materials.

The present work focuses on the use of a one-step pyrolysis method based on the thermal decomposition of TiOPC to synthesize a TiN–carbon composite. For comparative purposes, TiN was also prepared using an alkoxide method with urea as the nitrogen source to verify the synthesis. The structure and morphology of the resultant materials were characterized by XRD, FTIR, SEM, TEM, EDS, and XPS. The TiN–carbon composite materials were then used as anodes in LIBs, and their electrochemical performance was evaluated by CV and galvanostatic charge-discharge measurements.

## 2. Methods

### 2.1. Materials Synthesis

All the reagents were of analytical grade and used without further purification. The TiOPC precursor for the generation of the TiN–carbon composite (TiN(TD)) was prepared by refluxing TiCl_4_ with phthalonitrile in a 4:1 ratio in 1-chloronaphthalene for 6 h [31]. After the reaction, the mixture was cooled to room temperature and the product was obtained by vacuum filtration. The sample was washed with methanol and acetone and further purified by sublimation. After purification and drying, the TiOPC was placed in an alumina crucible and loaded into a quartz tube in a Thermolyne horizontal tube furnace (model F79330-33-70) (Thermo Fisher Scientific, Waltham, MA, USA). The quartz tube was sealed and purged with nitrogen (UHP) for 15 min. The temperature of the furnace was increased from room temperature to 750 °C at a rate of 10 °C min^−1^ and maintained at 750 °C for 5 h. After the reaction, the samples were allowed to cool to room temperature while maintaining the nitrogen flow. For comparison purposes, the H_2_PC was also synthesized using the same reaction conditions, without the presence of TiCl_4_, and then was carbonized using the same conditions to prepare carbon (H_2_PC(TD)). The TiN was prepared using a sol-gel method (TiN(SG)) [16]. In brief, 2.25 g of urea was dissolved in 5 mL of ethanol, and 1 mL of TiCl_4_ was added slowly under magnetic stirring. The resultant gel was poured into an alumina crucible and placed in a quartz tube in a horizontal tube furnace. The quartz tube was sealed and purged with nitrogen for 15 min. The temperature of the furnace was increased from room temperature to 750 °C at a rate of 10 °C min^−1^ and then maintained for 5 h. After 5 h of reaction, the sample was cooled to room temperature naturally while maintaining the nitrogen flow.

### 2.2. Characterization

The synthesized compounds were characterized by FTIR, XRD, SEM, TEM, and XPS. FTIR spectra of the dried and ground species were collected using a Perkin-Elmer Frontier FTIR spectrometer equipped with an attenuated total reflection (ATR) accessory. The spectra were collected over the range of 4000–650 cm^−1^, collecting 32 scans at a resolution of 4 cm^−1^. The data were analyzed using Spectrum software (Version 8.0, Perkin Elmer, Waltham, MA, USA). The XRD data were collected using a Bruker D2 phaser diffractometer with a cobalt source (K_α_ = 1.789 Å) and an iron filter. The materials were prepared by homogenizing samples into fine powders using a mortar and pestle. The patterns were collected from 10 to 80° 2θ with a 0.05° step size and 5 s counting time. The collected diffraction patterns were fitted using the Le Bail fitting procedure in the FullProf Suit software (Version 5.10) and crystallographic data from the literature [32,33,34,35].

The product morphology was observed using SEM. The SEM images were collected using a Sigma VP Carl Zeiss scanning electron microscope (Carl Zeiss, White Plains, NY, USA). The SEM was operated with accelerating voltages between 2.0 and 6.0 kV at a working distance of up to 6.5 mm. The SEM was equipped with an EDS from EDAX model Octane Super (EDAX, Pleasanton, CA, USA) and was used to determine the elemental distribution of C, O, N, and Ti in the synthesized samples. TEM images were taken using a FEI Titan G2 60-300 (Thermo Fisher Scientific, Waltham, MA, USA) microscope operated with an accelerating voltage of 200 kV. The samples were further analyzed using XPS to determine the surface chemistry of the samples. The XPS data were collected using a Thermo Scientific K-Alpha Photoelectron Spectrometer (Thermo Fisher Scientific, Waltham, MA, USA) and analyzed using the CASA XPS software (Version 2.3.25, Casa Software limited, Teignmouth, UK) [36]. The operation parameters were as follows: a micro-fused monochromatic Al K-α (1486.7 eV) source with scans at 0.1 eV and a spot size of 400 µm.

The CV of the anodes was evaluated using Li-ion half-cells (CR2032 coin cells, PHD Energy Inc., Georgetown, TX, USA). The electrodes were fabricated through a slurry-coating process. The slurry was prepared by mixing 90% of the active material and 10% (mass/mass) polyacrylonitrile (PAN) as the binder in dimethylformamide. A thin film of the mixed slurry was applied onto a 0.025 m thick copper foil and placed in a vacuum oven to be dried at 60 °C for 24 h. The dried coated copper foil was placed in an argon gas-filled tube furnace and heated to 450 °C for 5 h to pyrolyze the PAN binder. The electrodes were punched into 0.5″ diameter discs using a precision punch (Nagomi). Lithium metal was used as the counter electrode (0.38 mm thick, Sigma Aldrich, Rockville, MD, USA) with a glass-fiber separator (Separion S240 P25, 25 μm thickness) (Evonik Industries, Austin, TX, USA). The electrolyte used was 1 M LiPF_6_ dissolved in 1:1 (*v*/*v*) ethylene carbonate/dimethyl carbonate. The cells were assembled in a glovebox (Mbraun, Stratham, NH, USA) under a high-purity argon atmosphere (water and oxygen level < 0.5 ppm). CV experiments were performed at a scan rate of 0.2 mV s^−1^ over a voltage range between 0.05 and 3.0 V (Biologic Science Instruments, Seyssinet-Pariset, France). The electrochemical performance of the electrodes was evaluated by galvanostatic charge-discharge experiments using a LANHE battery testing system (CT2001A) with an applied current density of 100 mA g^−1^ over 100 cycles over the potential range of 0.05–3.0 V.

## 3. Results and Discussion

### 3.1. FTIR

Figure 1A,B shows the FTIR spectra collected for the TiOPC and metal-free phthalocyanine (H_2_PC) compounds, while Table 1 illustrates the identified FTIR bands. Figure 1B shows an expanded view of the FTIR from 1700 to 650 cm^−1^.

The FTIR spectra of the synthesized compounds showed characteristic bands to confirm the H_2_PC and TiOPC. The FTIR results confirmed the synthesis of TiOPC, where the bands at 798 cm^−1^ and 961 cm^−1^ indicated the presence of N-Ti-N [38] and Ti=O, respectively [37,40,41]. The N-H stretches observed at 3282 cm^−1^ in H_2_PC were absent in the TiOPC complex, indicating the loss of the N-H and the formation of the Ti-N bonds. In addition, the C-N stretches in TiOPC were shifted to a lower wavenumber than observed for H_2_PC, indicating a change in the coordination environment and the formation of Ti-N bonds in the complex [37,38].

### 3.2. SEM/EDS

SEM was used to study the morphology of synthesized materials. Figure 2A shows the SEM images of TiN synthesized by the sol-gel method (TiN(SG)), with an appearance of homogeneously stacked particles. The SEM image of TiN(TD) in Figure 2B indicates the presence of numerous rough-texture spheres of TiN evenly distributed throughout a tightly packed carbon material.

### 3.3. TEM

Figure 3 shows high-resolution TEM images of TiN(TD). Figure 3A shows that the TiN appears in small clusters throughout the carbon matrix, while Figure 3B reveals a uniform distribution of the TiN nanocrystals within the clusters. The average grain size was determined to be between 3–4 nm.

### 3.4. XRD

#### Titanium Compounds

The diffraction patterns of TiOPC, TiN(SG), and TiN(TD) are shown in Figure 4. Table 2 shows the Le Bail fitting of the titanium compounds [35]. It should be noted in the Le Bail fitting that the values of χ^2^ are below 5, indicating very good fits to the experimental data.

The results of the TiOPC fitting showed that the crystal was in a P-1 lattice, which was consistent with the space group identified in the literature [32]. It can be seen in Table 2 that the calculated lattice parameters were similar to those reported [33]. The lattice parameters of the TiN(SG) prepared using alkoxide-urea confirmed the formation of TiN. More importantly, the crystal lattice for TiN was determined to be FM3-M, a cubic lattice with *a* = *b* = *c* = 4.240 Å and χ^2^ = 1.50, in close agreement with literature values [34]. TiN(TD) also showed a cubic lattice FM3-M, with *a* = *b* = *c* = 4.240 Å and χ^2^ = 1.05, also in excellent agreement with results reported for TiN [34]. TiN(TD) showed the presence of amorphous carbon; the 002 peak for amorphous carbon was observed in the diffraction pattern around 2θ = 28°. The XRD results for both TiN(SG) and TiN(TD) showed the presence of the 111, 200, and 220 diffraction planes at 42.920, 49.978, and 73.372° in 2θ, respectively. The average grain size, determined using Scherrer’s equation and corrected for instrumental line broadening, was 7.97 ± 0.4 nm for TiN(SG) and 4.3 ± 0.9 for TiN(TD). The average grain size of TiN(TD) agreed with the values obtained from the TEM results.

### 3.5. XPS

Figure 5, Figure 6 and Figure 7 show high-resolution XPS spectra for TiN(SG), TiOPC, and TiN(TD), respectively. Table 3 shows the composition of the samples as determined from the XPS analysis based on atomic percentages. As can be seen in Table 3, the composition of all samples shows a low percentage of titanium. However, the percentage composition of TiN(TD) is approximately twice that observed in the TiOPC sample, which was expected due to the loss of hydrogen and carbon from the material during the carbonization process. Also, the amount of oxygen present in the sample increased, which is more than likely due to the adsorption of oxygen at the surface of the material. The amount of O_2_ present in TiN(SG) was also high, which is likely due to surface adsorption of O_2_ by the material.

Figure 5A shows the Ti 2p XPS data for TiN(SG), indicating the existence of Ti 2p_3/2_ and Ti 2p_1/2_ regions. The Ti 2p_3/2_ region was deconvolved into three components at 455.3, 456.7, and 458.4 eV, corresponding to Ti^3+^-N, Ti^3+^-N-O, and Ti^4+^-O binding environments, respectively [43,44,45]. The results indicate that the surface of the TiN has been oxidized. The surface oxidation of TiN(SG) was expected due to the small size of the TiN nanoparticles, determined by XRD to be about 8 nm and calculated based on the average of the peak width of TiN diffraction. Small nanoparticles are typically reactive and undergo surface oxidation readily. However, oxidation was not too extensive, as it was not apparent in the XRD. The Ti 2p_1/2_ portion of the spectrum had low intensity and was deconvolved into two peaks at 461.3 eV and 463.9 eV, corresponding to the Ti-N and Ti-O binding environments, respectively [46]. In addition, the peak representing the Ti-N-O binding environment was not observed. Figure 6A represents the Ti 2p_3/2_ and Ti 2p_1/2_ XPS of the TiOPC complex. A single peak was identified in the Ti 2p_3/2_ region at 457.2 eV, while the Ti 2p_1/2_ region exhibited a peak at 463.1 eV. This alignment is consistent with the presence of Ti^4+^, indicating the presence of TiOPC, as reported in the literature [47,48]. Figure 7A shows the Ti 2p spectrum obtained for the TiN(TD) complex. The Ti 2p_3/2_ signal was deconvolved into three peaks at 456.7, 458.1, and 459.1 eV, identified as the Ti^3+^-N, Ti^3+^-N-O, and Ti^4+^-O binding environments, respectively [45,49]. The binding energies were shifted slightly due to the particle size of 4.3 nm. The shift in the peak positions to higher energies was observed for nanocrystals due to surface strain and surface defects [50,51]. The Ti 2p_1/2_ signal was deconvolved into two peaks at 462.9 and 464.2, indicating the presence of TiN and TiO, respectively [52].

Figure 5B shows the C 1s XPS spectrum collected from TiN(SG). Only one region was present and deconvolved into three peaks at 284.8, 286.1, and 288.5 eV, corresponding to C-C, C-O, or C-N, and the C satellite peak, respectively [42]. Figure 6B shows the C 1s spectra for the TiOPC complex. The C 1s spectrum was deconvolved into four peaks at binding energies of 283.7, 284.9, and 287.4 eV, corresponding to the C-C, C=C, and C-N bonds, respectively [53]. Figure 7B shows the C 1s spectra for TiN(TD). The sample was deconvolved into three peaks located at binding energies of 284.5, 285.7, and 288.4, corresponding to C=C, C-C, and a satellite peak, respectively [42,54]. There is no indication of the formation of Ti-C, which would be located at a binding energy of 281.8 eV [55], indicating that the sample is a mixture of TiN and carbon. Neither of the synthesized TiN samples showed the presence of TiC, indicating no TiC formed in the conversion process [56].

Figure 5C shows the N 1s region for TiN(SG). The N 1 spectrum consists of four peaks located at 396.1, 397.1, 398.8, and 400.8, identified as N-O-Ti, N-Ti, N-O, and N-O, respectively [48]. Figure 6C shows the N 1s spectrum for the TiOPC complex, which was deconvolved into two peaks at 397.9 and 399.3 eV, assigned to the N-Ti and N-C binding environments of phthalocyanine, respectively [57]. Figure 7C shows the N 1s XPS spectrum for TiN(TD). The XPS spectrum of the TiN(TD) sample was deconvolved into two peaks located at 398.2 eV and 400.5 eV. The two peaks observed in the N1s spectrum corresponded to the N-Ti and N-C binding environments [44,46].

Figure 5D shows the O 1s XPS spectrum for TiN(SG), which was deconvolved into two individual peaks at 529.9 and 531.7 eV, determined to be the O_2_ adsorbed and the O-N-Ti oxygen environments, respectively [46,58]. These results are consistent with the binding environments observed in the Ti 2p for TiN(SG). Figure 6D shows the O 1s XPS spectra for the TiOPC complex, which was deconvolved into three peaks at 529.3, 531.4, and 532.9 eV, which were determined to be adsorbed oxygen, the oxygen bound to Ti (IV), indicating the formation of titanyl phthalocyanine (TiOPC), and the presence of water/hydroxide in the sample [58,59]. Figure 7D shows the O 1s spectrum of TiN(TD). The O 1s spectrum consisted of two peaks located at 529.9 and 532.2 eV, which were determined to be oxygen adsorbed and O-N-Ti and the presence of hydroxide or water in the sample [45,58]. The results of the XPS fittings are summarized in Table 4.

## 4. Cyclic Voltammetry

Cyclic voltammetry (CV) measurements of the initial three cycles for thermally decomposed H_2_PC (H_2_PC(TD)) and TiN(TD) are shown in Figure 8. The experimental data were collected over a potential window between 0.05 and 3.0 V (vs. Li^+^/Li) at a scan rate of 0.2 mVs^−1^. Figure 9A shows a difference between the first and second cycles due to the formation of the solid electrolyte interface (SEI) layer at the first cathodic scan (lithiation). The second and subsequent cycles overlap, indicating good reversibility and high coulombic efficiency. The anodic peaks observed at 2.5 V correspond to the pyrolysis of the polyacrylonitrile binder during the heat treatment of the slurry [60]. The thermal decomposition of H_2_PC and thermal treatment performed on its slurry resulted in the formation of a carbon electrode. The peak around 0.5 V observed on the first cathodic scan indicates the formation of the SEI layer, while the anodic peaks around 1 V indicate the intercalation between Li^+^ and carbon.

TiN(TD), depicted in Figure 8B, exhibits a difference between the first and second cycles, which is also attributed to the SEI formation. The small cathodic peak around 0.6 V is indicative of the SEI layer formation at the first discharge process. Multiple cathodic peaks were observed in the first discharge process (lithiation), indicating a complex reaction between TiN and Li. It is also worthwhile to mention that the electrode contained a substantial amount of carbon because of the synthesis method and thermal treatment process performed on the starting materials (slurry), which may result in a complicated CV profile. The cathodic and anodic peaks around 0.5 V are assigned to the lithiation and delithiation of the carbon phase in the electrode, respectively [61].

Figure 9A,B shows the galvanostatic charge/discharge curves of H_2_PC(TD) and TiN(TD) anodes after 100 cycles at a current density of 100 mAg^−1^ within a voltage window of 0.1–3 V. Both anodes show high irreversible capacities at the first discharge (Li-insertion) cycle. The irreversible capacity may be caused by the reductive decomposition of electrolyte solution and the subsequent formation of the SEI layer at the anode surface. However, the TiN(TD) anode showed a lower irreversible capacity at the first charge cycle (44% loss in capacity) than that observed for the H_2_PC(TD) anode (58% loss). The capacity retention after the second cycle for both electrodes was good (almost 100%), resulting in a high coulombic efficiency of nearly 100%, indicating the formation of a stable interface (SEI layer) during subsequent charge-discharge cycles.

Figure 10 shows the cycle performance and coulombic efficiency of H_2_PC(TD) and TiN(TD) electrodes after 100 cycles at 100 mAg^−1^. The H_2_PC(TD) anode exhibited a stable charge capacity, which is consistent with the observation that the H_2_PC(TD) anode exhibits a higher theoretical capacity than TiN(TD). After the fifth cycle, the capacity of the H_2_PC(TD) anode started to increase, and after 30 cycles, the capacity was decreased to 420 mAhg^−1^ and remained stable up to 100 cycles. On the other hand, the charge capacity of the TiN(TD) electrode decreased after a few cycles and then remained constant at 350 mAhg^−1^ after 100 cycles, indicating good electrochemical stability. The TiN(TD) anode exhibits higher initial coulombic efficiency (56.1%) and better capacity retention (98.3% of the second cycle) than the H_2_PC(TD) anode. The improvement of TiN(TD) capacity compared to commercial graphite anode or carbon fiber anodes might be due to the synergistic effect of TiN on the electrochemical performance of the carbon matrix anode and might also be due to the good electrical conductivity and high chemical and thermal stability of TiN. However, pure TiN suffers from irreversible oxidation reactions, which lead to poor cyclability [62]. For this reason, the addition of conductive carbon to TiN can prevent the oxidation reaction and improve the cyclability of the TiN anode [62]. In fact, there is a contribution from TiN and carbon to the capacity observed for the TiN(TD) anode. Previous experiments performed on carbon-fiber anodes under the same conditions (current density of 100 mAhg^−1^) have shown the capacity of the carbon-fiber anode was approximately 200 mAhg^−1^ after 100 cycles, which is lower than those observed in the present work [63,64,65]. TiN has been used as a coating material for high-capacity anodes in LIBs such as Si and SnO_2_ [66,67]. The TiN/Si and TiN/SnO_2_ composite anodes exhibited good electrochemical performance compared to the Si and SnO_2_ anodes. This was in part due to the good electrical conductivity and chemical and thermal stability of TiN [66,67].

## 5. Conclusions

The thermal decomposition of TiOPC turned out to be a facile, rapid, and effective process for preparing anode materials for LIBs, offering the advantages of using precursors fabricated by well-known compounds with a long shelf life and good chemical stability. In addition, the material showed the presence of small TiN nanoparticles approximately 3–4 nm in size attached to the carbon matrix. The nanoparticles were present in clusters of small spherical particles observed throughout the material. The composition of the nanoparticles was confirmed by XPS and XRD, which showed the presence of TiN and no evidence of the synthesis of TiC (as a by-product). The electrochemical performance showed both the carbon synthesized from the H_2_PC and the TiN(TD) materials had high cycle stability and excellent electrochemical performance with a stable capacity over 100 cycles at 100 mAg^−1^. The potential exists for a carbon-stabilized TiN material to be easily synthesized for LIB anode applications.

## Figures and Tables

**Figure 1 nanomaterials-14-00624-f001:**
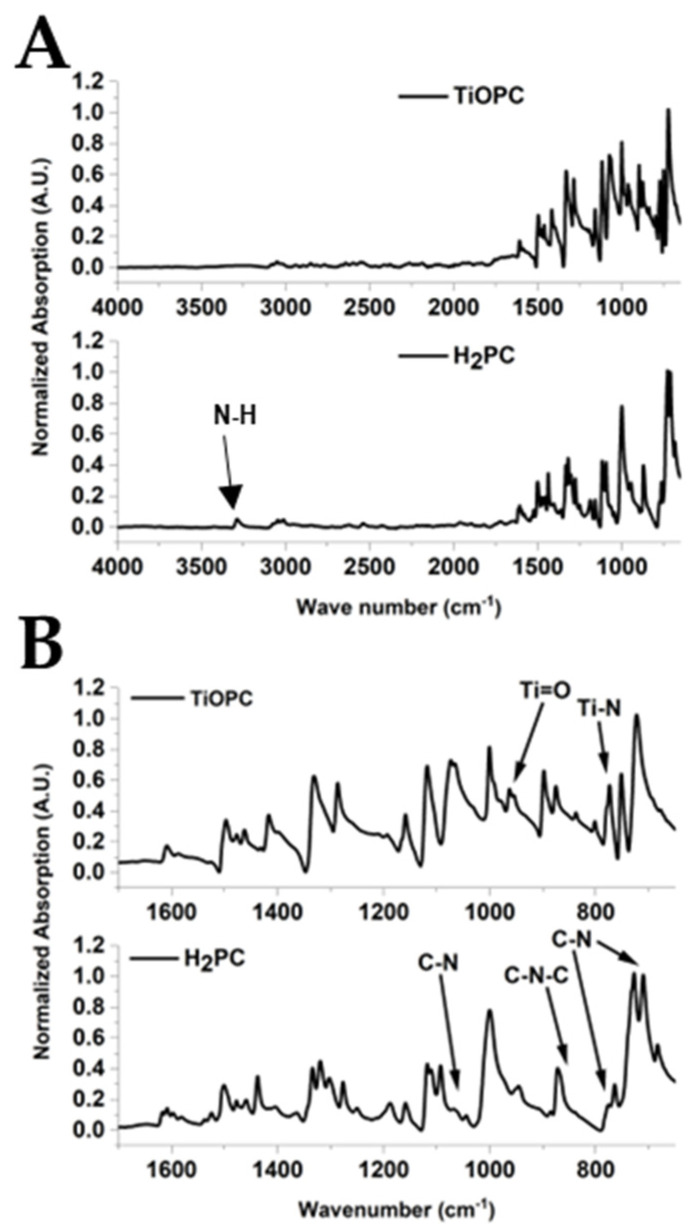
(**A**) FTIR spectra of the synthesized TiOPC and H_2_PC starting materials. (**B**) Expanded view of FTIR from 1700 to 650 cm^−1^.

**Figure 2 nanomaterials-14-00624-f002:**
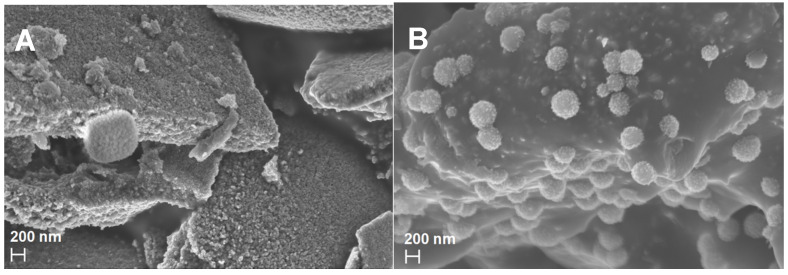
SEM images of (**A**) TiN(SG) and (**B**) TiN(TD).

**Figure 3 nanomaterials-14-00624-f003:**
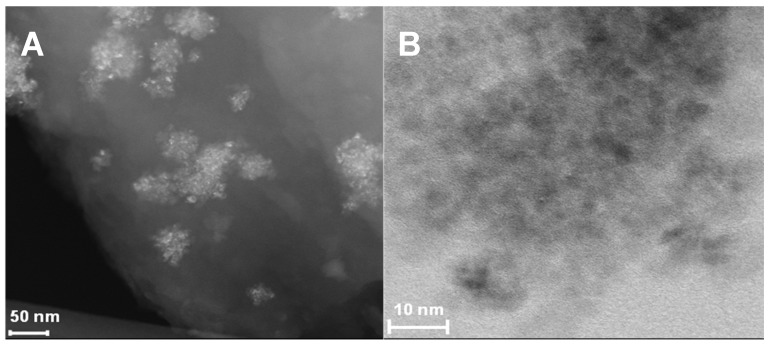
High-resolution TEM images of TiN(TD).

**Figure 4 nanomaterials-14-00624-f004:**
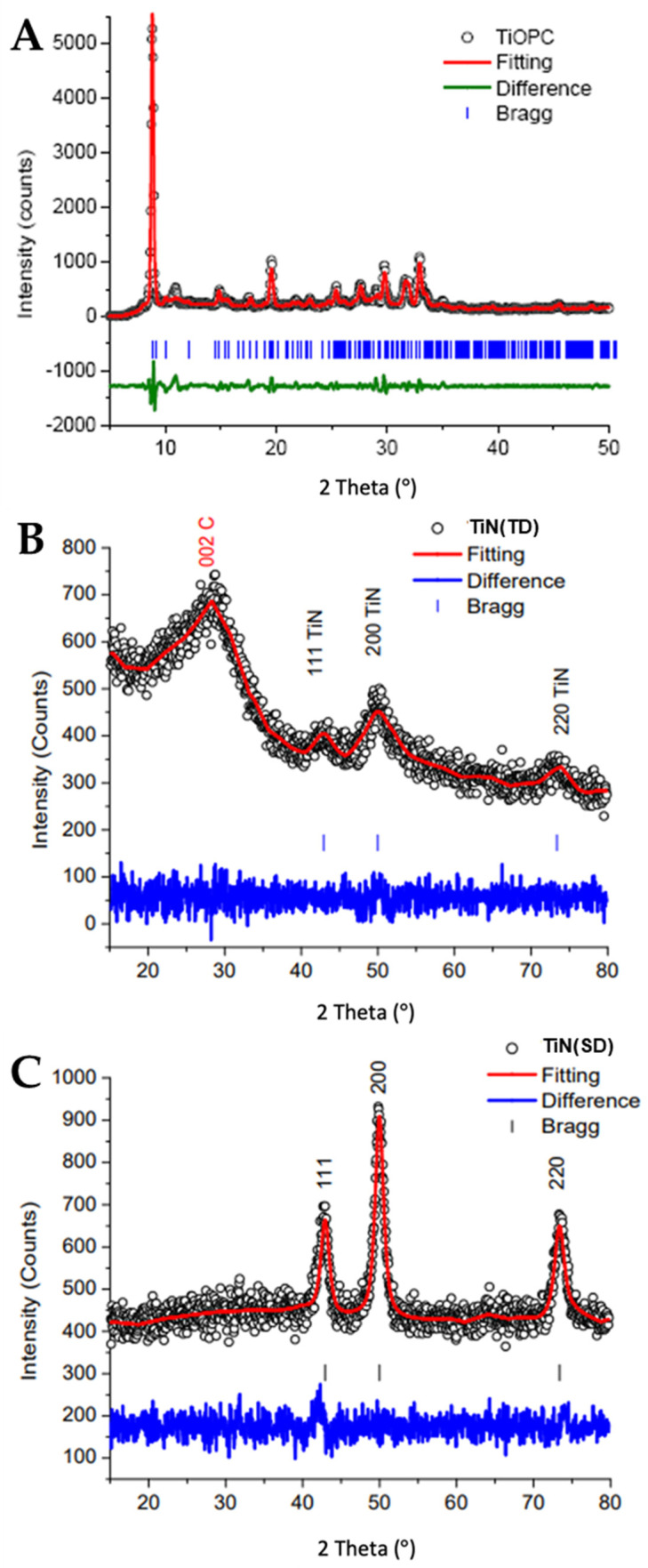
XRD patterns for (**A**) TiOPC, (**B**) TiN(TD), and (**C**) TiN(SG).

**Figure 5 nanomaterials-14-00624-f005:**
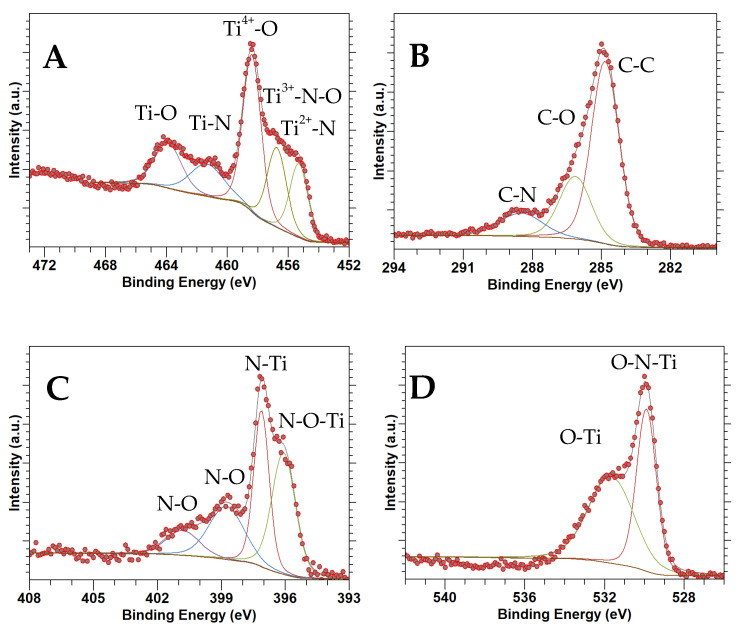
XPS spectra of TiN(SG): (**A**) Ti 2p, (**B**) C 1s, (**C**) N 1s, (**D**) O 1s.

**Figure 6 nanomaterials-14-00624-f006:**
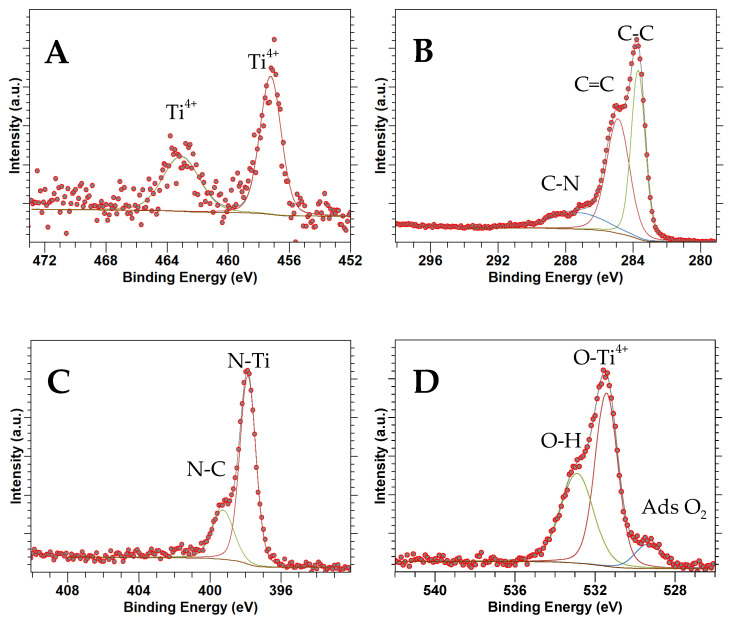
XPS spectra of TiOPC: (**A**) Ti 2p, (**B**) C 1s, (**C**) N 1s, (**D**) O 1s.

**Figure 7 nanomaterials-14-00624-f007:**
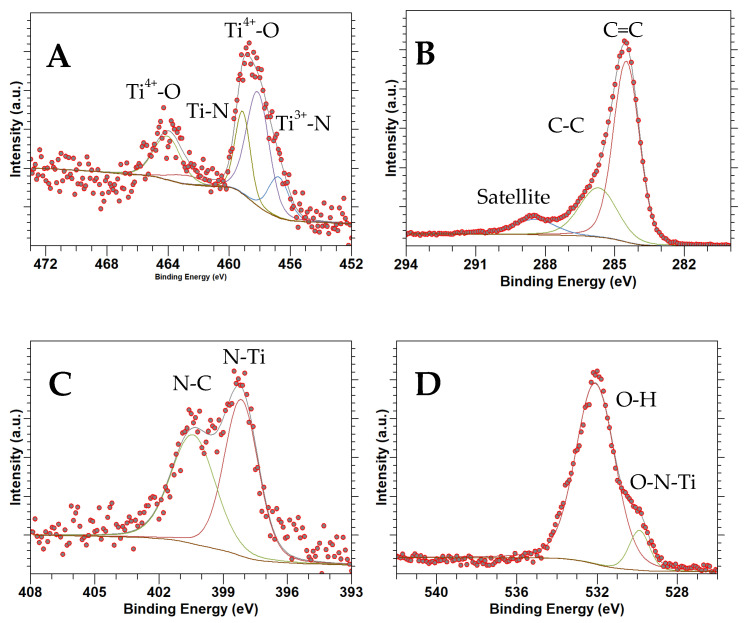
XPS spectra of TiN(TD): (**A**) Ti 2p, (**B**) C 1s, (**C**) N 1s, (**D**) O 1s.

**Figure 8 nanomaterials-14-00624-f008:**
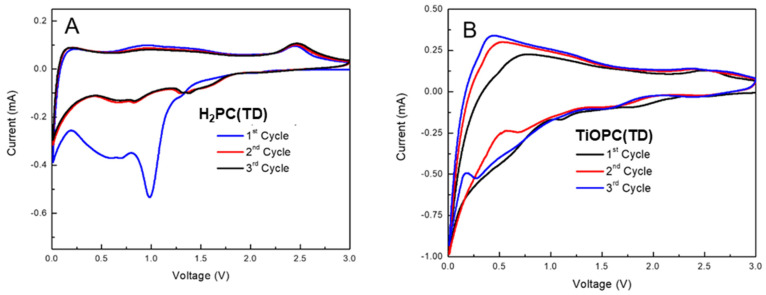
Cyclic voltammetry curves of (**A**) H_2_PC(TD) and (**B**) TiN(TD) electrode materials.

**Figure 9 nanomaterials-14-00624-f009:**
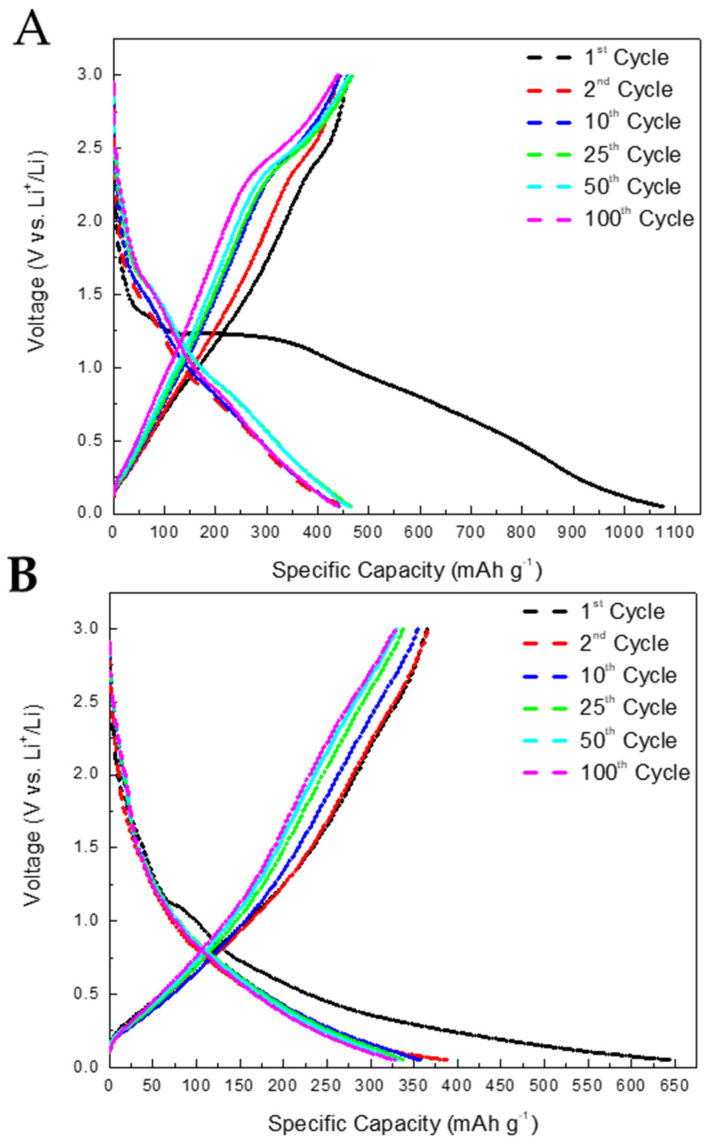
(**A**) Charge/discharge curves at a current density of 100 mAg^−1^ H_2_PC(TD). (**B**) Charge/discharge curves of TiN(TD) anode at a current density of 100 mAg^−1^.

**Figure 10 nanomaterials-14-00624-f010:**
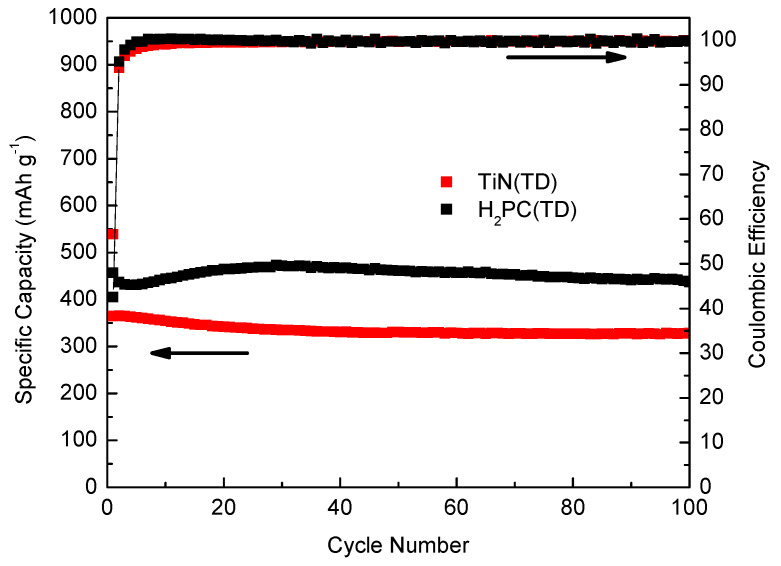
Electrochemical performance and coulombic efficiency of thermally carbonized metal-free phthalocyanine and titanium phthalocyanine electrode materials.

**Table 1 nanomaterials-14-00624-t001:** FTIR band position and respective assignments for the H_2_PC and TiOPC.

H_2_PC Band Position (cm^−1^)	TiOPC Band Position (cm^−1^)	Assignment
710		C-N [37]
729	724	C-H out of plane deformation [38,39]
762	751	Macrocycle ring stretching [c] [37]
778	778	C-N stretching [38]
	798	N-Ti stretching [37]
839	832	C-N-C ring breathing
872	877	N-H stretching coupling with isoindole deformation [37]
	889	Isoindole deformation with coupling aza stretching [37]
944		
	961	Ti=O [31,37,40]
998	1002	Benzene ring and C=C [37]
1064	1062	C–N stretching in pyrrole vibration [38]
1075		
1091		
1116	1117	C–H in-plane deformation [37,38,39]
1157	1158	C–N in-plane and C–H in-plane [38,41]
1187		Isoindole stretching [38]
1275		
1299	1286	C–N in isoindole stretching [37,38]
1324		
1336	1330	C–C in isoindole [38]
1417	1413	Isoindole stretching [37]
1437		
1461	1460	C–H in-plane bending [38]
1477	1474	C=N pyrrole [37]
1501	1489	C–H bending in aryl [38]
1523		C–H aryl [38]
1576		
1595	1585	Benzene C-C stretching [42]
1610	1609	C–C stretching vibration in pyrrole [38]
2923	2923	C–H stretching [38]
3004	3004	C-H stretching [38]
3050	3050	C–H stretching vibration in ring [38]
3282		N-H [38]

**Table 2 nanomaterials-14-00624-t002:** Le Bail fitting results for TiOPC, TiN(SG), and TiN(TD).

Compound	Space Group	a (Å)	*b* (Å)	c (Å)	α (°)	β (°)	γ (°)	* χ^2^	Reference
α-H_2_PC_syn_	C_2_/n	25.755	3.773	23.398	90	93.111	90	2.07	This Work
α-H_2_PC_lit_	C_2_/n	26.121	3.797	23.875	90	94.16	90	** N/A	[32]
TiOPC_syn_	P-1	12.269	12.598	8.594	95.067	96.127	67.818	3.74	This Work
TiOPC_lit_	P-1	12.17	12.58	8.64	95	96.3	67.9	N/A	[33]
TiN(TD)	FM3-M	4.24 (0)	4.24 (0)	4.24 (0)	90.0	90.0	90.0	1.05	This Work
TiN(SG)	FM3-M	4.24 (0)	4.24 (0)	4.24 (0)	90.0	90.0	90.0	1.50	This Work
TiN_lit_	FM3-M	4.244	4.244	4.244	90.0	90.0	90.0	N/A	[34]

Note: * is the Chi squared value which is an indicator of the goodness of fit and ** N/A means Not Applicable.

**Table 3 nanomaterials-14-00624-t003:** Composition of the synthesized nanomaterials as determined from XPS analysis of the samples.

Sample	%Ti	%N	%C	%O
TiOPC	0.6	11.3	81.7	6.4
TiN(TD)	1.3	5.0	82.5	11.3
TiN(SG)	16.1	12.5	52.2	19.2

**Table 4 nanomaterials-14-00624-t004:** A summary of the XPS fitting results for each of the elements observed in the samples.

Sample	Ti 2p_3/2_	Energy (eV)	Ti2P_1/2_	O 1s	Energy (eV)	N1S	Energy (eV)	C 1s	Energy (eV)
TiOPC	Ti^4+^-N/O	457.2	463.1	O-Ti	529.3	N-Ti	397.9	C-C	283.7
				O-H	531.4	N-C	399.3	C=C	284.9
				O_2ads_	532.9			C-N	287.4
TiN(TD)	Ti-N	456.7	462.9	O-N-Ti	529.9	N-Ti	398.2	C-C	284.5
	Ti-N-O	458.1		O-H	532.2	N-C	400.5	C-O/C-N	285.7
	Ti-O	459.1	464.2					Satellite	288.4
TiN(SG)	Ti-N	455.5	461.3	O_2ads_	529.9	N-O-Ti	396.1	C-C	284.8
	Ti-N-O	456.7		O-N-Ti	531.7	N-Ti	397.1	C-O/C-N	286.1
	Ti-O	458.4	469.3			N-O	398.8	Satellite	288.4
						N=O	400.8		

## Data Availability

All the relevant data that support the findings of this study are available from the corresponding authors on reasonable request.

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
