# Peer review of "Synthesis and Characterization of Titanium Nitride–Carbon Composites and Their Use in Lithium-Ion Batteries"

_nanomaterials, 2024, doi:10.3390/nano14070624_

Round 1

Reviewer 1 Report

Comments and Suggestions for Authors

In this manuscript, authors synthesized TiN-carbon composite by thermal decomposte synthesized TiOPC or solgel method. Chemical and electrochemical properties were charactrized for the synthesized materials.Some revision are suggested for paper publication.

1. Can author explain why figure 2 and figure 3 shows so different in mophology? can you present the SEM image in different maginication?

2.For XPS data, please provide the relative  atomic ratio for Ti, N, C, O. Only XPS spectra is not enough to support. 

3.For a electrochemical data, is there any full cell data? Can author provide the voltage profile for cells in figure 10?

4. Wrong legend for Figure 5c. 

Author Response

Response to Reviewer: We would like to thank the reviewers for their feedback and comments on the manuscript. Below are our responses to the reviewer’ comments on the manuscript in a comment- response dialogue.

Reviewer#1

In this manuscript, authors synthesized TiN-carbon composite by thermal decomposte synthesized TiOPC or solgel method. Chemical and electrochemical properties were charactrized for the synthesized materials. Some revision are suggested for paper publication.

Comment: Can author explain why figure 2 and figure 3 shows so different in mophology? can you present the SEM image in different maginication?

Response: Due to the difference in the magnification of the two figures, we have decided to remove Figure 3 since it is not overly necessary for the manuscript. The SEM in our lab has been down waiting on parts, hence no more imaging can be taken at this time since we need to meet the deadline for the submission of the revised manuscript.

Comment For XPS data, please provide the relative atomic ratio for Ti, N, C, O. Only XPS spectra is not enough to support. 

Response: We have added Table 3 to the revised manuscript to show the elemental composition of the synthesized materials.

Table 3: Composition of the synthesized nanomaterials as determined from XPS analysis of the samples.

Sample

%Ti

%N

%C

%O

TiOPC

0.6

11.3

81.7

6.4

TiN(TD)

1.3

5.0

82.5

11.3

TiN(SG)

16.1

12.5

52.2

19.2

We have also added the following paragraph to the revised manuscript:

Table 3 shows the composition of the samples as determined from the XPS analysis, based on atomic percentages. As can be seen in Table 3, the composition of all samples shows a low percentage of titanium. However, the percentage composition of the TiN(TD) is approximately twice than that observed in the TiOPC sample, which was expected due to the loss of hydrogen and carbon from the material during the carbonization process. Also, the amount of oxygen present in the sample increased, which is more than likely due to the adsorption of oxygen at the surface of the material. The amount of O2 present in the TiN(SG) was also high, which is likely due to surface adsorption of O2 by the material. 

Comment. For a electrochemical data, is there any full cell data?

Response: The data on full cells are not available currently since it takes a long time to synthesize, characterize and test the electrode materials in full cells. In fact, performing the electrochemical experiments is time consuming and we have a time restriction to submit the revised version of the manuscript. In the present work, the electrochemical performance of the TiN(TD) and H2PC(TD) anodes was evaluated using Li-ion half cells since the focus of the manuscript is on the synthesis, characterization, and the electrochemical performance of one single electrode material (anode) and it is more valuable to discuss the results of one electrode rather than on 2 electrodes (i.e., anode and cathode) using Li-ion full cells.  

Comment: Can author provide the voltage profile for cells in figure 10?

Response: The voltage profile of the anodes associated with the results of Figure 9 is shown in Figure 9 (a and b). The following paragraph and Figures were added to the revised manuscript.

Figure 9 (a and b) shows the galvanostatic charge/discharge curves of TiN(TD) and H2PC(TD)  anodes after 100 cycles at a current density of 100 mAg-1 within a voltage window of 0.1-3 V. Both anodes show high irreversible capacities at the first discharge (Li-insertion) cycle. The irreversible capacity may be caused by the reductive decomposition of electrolyte solution and the subsequent formation of the SEI layer at the anode surface. However, the TiN(TD) anode showed a lower irreversible capacity at the first charge cycle (44% loss in capacity) than that for the H2PC(TD) anode (58% loss). The capacity retention after the second cycle for both electrodes was good (almost 100 %) resulting in a high coulombic efficiency of nearly 100% indicating the formation of a stable interface (SEI layer) during subsequent charge/discharge cycles.

There is a new Figure 9 in the revised manuscript.

Figure 9A. Charge/discharge curves at a current density of 100 mAg-1  H2PC(TD). B. Charge/discharge curves of TiPC(TD) anode at a current density of 100 mAg-1.

Comment  Wrong legend for Figure 5c. 

Response: This has been corrected to read: TiN(SG)

Reviewer 2 Report

Comments and Suggestions for Authors

The manuscript nanomaterials-2931862 by Helia Magali Morales et al. proposes the preparation of TiN material through two different synthetic routes, the physicochemical characterization of the materials and their application into Li-ion batteries.

However, there is a disappointing discrepancy between the characterization of the materials and the electrochemical testing, where TiN(SG) is completely missing and H2PC(TD) material is introduced for the first time and apparently works better than TiN(TD).

Which is the advantage of TiN(TD) over H2PC(TD)? Nevertheless, I would expect that H2PC(TD) is even cheaper.

Indeed, in the present version, the take-home message of the work is not clear, despite I can appreciate the complete investigation based on multiple techniques.

This aspect should be clarified, before that the manuscript can be considered suitable for publication.

Moreover, hereafter there are other minor issues to be addressed in order to improve the quality of the work.

In the section “2.1 Materials Synthesis” I would clearly separate the procedure for the synthesis of TiN(TD) and the one for TiN(SG). As a matter of fact, TiOPC is used as precursor only for TiN(TD), but it is not clearly indicated at the beginning of the paragraph.

Section 3.1 reports the FTIR analysis to monitor the conversion of phthalocyanine into TiOPC. The analysis is scientifically rigorous and the assignment of the bands is really precise, but the whole section is a little bit off-topic, considering that it is only referred to the precursor of TiN(TD) and that the following sections instead deal with the comparison between TiN(TD) and TiN(SG). Therefore, I would suggest to move section 3.1 in Supporting Information. Moreover, I would recommend to add in Figure 1 a magnification of the range 1700-700 cm-1, which is discussed in details in the table but is hardly visible in the figure.

Figure 3 does not allow a correct comparison between the two materials since the corresponding pictures have a quite different magnification scale (ten times larger in one case with respect to the other).

I would say that the discussion in the text of Figures 4A and 4B is reversed. Furthermore, an average grain of “3.3 ± 0.2 nm” estimated from TEM pictures denotes an accuracy that cannot be achieved by that technique, I would simply say “3-4 nm”.

In Figure 5, the label inside part C should be “SG” (and not “TD”).

I would suggest to resume the XPS results of Figures 6, 7, and 8 into a Table, in order to easily compare the relative proportions of the components in the different materials.

SEI is “interface” and not “interphase”.

The theoretical capacity of hard carbon in Li-ion batteries is 350 mAh/g, the capacity of graphite is 370 mAh/g, how is possible that the capacities reported in Figure 10 are higher? At least, to validate the results I would test the performances of commercial graphite in the same conditions.

The Coulombic efficiency should be reported together with the capacity values.

Author Response

Response to Reviewer: We would like to thank the reviewers for their feedback and comments on the manuscript. Below are our responses to the reviewer’ comments on the manuscript in a comment- response dialogue.

Reviewer#2

The manuscript nanomaterials-2931862 by Helia Magali Morales et al. proposes the preparation of TiN material through two different synthetic routes, the physicochemical characterization of the materials and their application into Li-ion batteries.

Comment: However, there is a disappointing discrepancy between the characterization of the materials and the electrochemical testing, where TiN(SG) is completely missing and H2PC(TD) material is introduced for the first time and apparently works better than TiN(TD).

Response: the focus of the present manuscript is on the synthesis and characterization of titanium nitride by the thermal decomposition of titanyl phthalocyanine (TiN (TD)) precursor and not on the TiN synthesis by sol-gel. For comparison purposes, we performed structure characterization by SEM and XRD on TiN prepared by sol-gel method. In fact, TiN has been recently used as coating materials with high-capacity anodes such as Si and SnO2 to improve their electrochemical performance and reduce the volume change after prolonged charge/discharge cycles [Electrochem. Solid-State Lett., 2000, 3, 493 and New J. Chem., 2013, 37, 2096.] The characterization of the H2PC(TD) is not feasible, FTIR proves to not be very informative with respect to the analysis of carbon-black or graphitic types of carbons. Also, the XRD of the sample would show a large broad peak around 30 in 2 theta which would not provide much insight to the sample or its structure. This peak was observed in the TiN(TD) sample. Also, the XPS provided a C1s spectrum which looked very similar to the C1s of the TiN(TD). In fact,the main thrust of the present manuscript is the synthesis and electrochemical performance of TIN (TD) by thermal decomposition to prepare a TiN-C composite material.

Comment: Which is the advantage of TiN(TD) over H2PC(TD)? Nevertheless, I would expect that H2PC(TD) is even cheaper.

Response: The cost can play an important role in the selection of the anode material. However, TiN is a better candidate than H2PC(TD) due to its good electrical conductivity (4000–55 500 S cm-1), and high chemical and thermal stability [/Electrochim. Acta, 2010, 55, 9024/ and, Nano Lett., 2012, 12, 5376; and also J. Mater. Chem. A, 2014, 2,10825]. Also, the capacity loss in the first cycle for the TiN(TD) was lower than that observed for H2PC(TD) anode, 44 and 58%, respectively.  It is cheaper to make the H2PC than the TiOPC, this is true; however, the amount of sample loss during pyrolysis is much larger for the H2PC than the TiOPC.  The H2PC is much more volatile than the TiOPC.  Thus, the conversion of H2PC to H2PC(TD) is less efficient and uses 3 to 4 times the amount of starting material. The loss during carbonization is double for H2PC compared to TiOPC, so in the end, TiOPC works out to be cheaper.

Comment: Indeed, in the present version, the take-home message of the work is not clear, despite I can appreciate the complete investigation based on multiple techniques.

This aspect should be clarified, before that the manuscript can be considered suitable for publication.

Response: there have been several papers recently published showing that TiN can be used to enhance the properties of carbon in lithium-ion battery anodes. The majority of these either synthesize the TiN on a carbon matrix to initiate a reaction between Ti and the carbon matrix then converts to a nitride.  However, in the current study, we show that it is possible to produce a TiN-carbon composite, that does not show the presence of TiC (which is a common contaminate and is somewhat detrimental to the behavior of the anode).  In addition, we have done this using a well-known and well used commonly synthesized material at relatively low temperature, low pressure, and good efficiency.

We have added the following paragraph to the introduction:

The main goal of the present work is the synthesis and characterization of TiN (TD) by the thermal decomposition of titanyl phthalocyanine precursor as well as the possibility of using TiN/C composite as anodes for LIBs. The current manuscript discusses in detail the synthesis and characterization of TiN as well as the synthesis of carbon from H2PC. Further work will be conducted to investigate the electrochemical performance of these materials.  

Minor issues to be addressed in order to improve the quality of the work.

Comment: In the section “2.1 Materials Synthesis” I would clearly separate the procedure for the synthesis of TiN(TD) and the one for TiN(SG). As a matter of fact, TiOPC is used as precursor only for TiN(TD), but it is not clearly indicated at the beginning of the paragraph.

Response:  The authors have addressed this issue according to the reviewer’s comment

We have revised the following paragraph to read as follows:

All the reagents were of analytical grade and used without further purification. The TiOPC precursor for the generation of the TiN-carbon composite (TiN(TD) was prepared by refluxing TiCl4 with phthalonitrile in a 4:1 ratio in 1-chloronaphthalene for 6 h [31]. After the reaction, the mixture was cooled to room temperature and the product was obtained by vacuum filtration. The sample was washed with methanol and acetone and further purified by sublimation. After purification and drying, the TiOPC was placed in an alumina crucible and loaded into a quartz tube in a Thermolyne horizontal tube furnace (model F79330-33-70). The quartz tube was sealed and purged with nitrogen (UHP) for 15 min. The temperature of the furnace was increased from room temperature to 750 °C at a rate of 10 °C min–1 and maintained at 750 °C for 5 h. After the reaction, the samples were allowed to cool to room temperature while maintaining the nitrogen flow. For comparison purposes, the H2PC was also synthesized using the same reaction conditions, without the presence of TiCl4, and then was carbonized using the same conditions to prepare carbon (H2PC(TD)).

Comment Section 3.1 reports the FTIR analysis to monitor the conversion of phthalocyanine into TiOPC. The analysis is scientifically rigorous and the assignment of the bands is really precise, but the whole section is a little bit off-topic, considering that it is only referred to the precursor of TiN(TD) and that the following sections instead deal with the comparison between TiN(TD) and TiN(SG). Therefore, I would suggest to move section 3.1 in Supporting Information. Moreover, I would recommend to add in Figure 1 a magnification of the range 1700-700 cm-1, which is discussed in details in the table but is hardly visible in the figure.

Response: The authors respectfully disagree with moving the section to the supplemental materials.  Because of the importance of reaction conditions and precursors on final products, we think that this section is very relevant to the manuscript.  However, we have added Figure 1 B to show a magnified section of the FTIR from 1700-700 cm-1 as requested by the reviewer.

We have added a revision of Figure 1 please see the text of the manusicript

We have revised the following paragraph /lines 177-181/ (lines 177-181 in the revised mansucript) with respect to Figure 1 to read: 

Figure 1 (A and B) shows the FTIR spectra collected for the TiOPC and metal-free phthalocyanine (H2PC) compounds while Table 1 illustrates the identified FTIR bands. Figure 1 B shows an expanded view of the FTIR from 1700-650 cm-1.

Comment: Figure 3 does not allow a correct comparison between the two materials since the corresponding pictures have a quite different magnification scale (ten times larger in one case with respect to the other).

Response: Due to the difference in the magnification between Figures 3a and 3b, we have decided to remove the figure and discussion on the EDS results since we cannot analyze more SEM images at this moment. 

Comment: I would say that the discussion in the text of Figures 4A and 4B is reversed (now Figure 3). Furthermore, an average grain of “3.3 ± 0.2 nm” estimated from TEM pictures denotes an accuracy that cannot be achieved by that technique, I would simply say “3-4 nm”.

Response:  We agree with the reviewer and have replotted Figure 4 and revised the following sentence in the manuscript to read: The average grain size was determined to be between 3-4 nm.

Comment: In Figure 5, the label inside part C should be “SG” (and not “TD”).

Response: this has been corrected in Figure 5, which is now Figure 4.

Comment: I would suggest to resume the XPS results of Figures 6, 7, and 8 into a Table, in order to easily compare the relative proportions of the components in the different materials.

Response: This has been corrected by adding Table 4 to the revised manuscript.

Table 4: Summarized results of XPS fitting for each of the elements observed in the samples.

Sample

Ti 2p3/2

Energy (eV)

Ti2P1/2

O1S

Energy (eV)

N1S

Energy (eV)

C1S

Energy (eV)

TiOPC

Ti4+-N/O

457.2

463.1

O-Ti

529.3

N-Ti

397.9

C-C

283.7

O-H

531.4

N-C

399.3

C=C

284.9

Ads O2

532.9

C-N

287.4

TiN(TD)

Ti-N

456.7

462.9

O-N-Ti

529.9

N-Ti

398.2

C-C

284.5

Ti-N-O

458.1

O-H

532.2

N-C

400.5

C-O/C-N

285.7

Ti-O

459.1

464.2

Satellite

288.4

TiN(SG)

Ti-N

455.5

461.3

O2ads

529.9

N-O-Ti

396.1

C-C

284.8

Ti-N-O

456.7

O-N-Ti

531.7

N-Ti

397.1

C-O/C-N

286.1

Ti-O

458.4

469.3

N-O

398.8

Satellite

288.4

N=O

400.8

The following sentence was added to the revised manuscript at line 325:

The results of the XPS fittings are summarized in Table 4..

Comment: SEI is “interface” and not “interphase”.

Response: This term was corrected in the revised manuscript. 

Comment: The theoretical capacity of hard carbon in Li-ion batteries is 350 mAh/g, the capacity of graphite is 370 mAh/g, how is possible that the capacities reported in Figure 10 are higher?

Response: This is a good point. The improvement of TiN/C capacity compared to commercial graphite, hard carbon or carbon-fiber anodes might be due the synergistic effect of TiN on the electrochemical performance of the carbon-matrix anode and might be also due to the good electrical conductivity and high chemical and thermal stability of TiN. However, pure TiN suffers from irreversible oxidation  reactions, which lead to poor cyclability [https://doi.org/10.1002/aenm.201502159]. For this reason, the addition of conductive carbon to TiN can prevent the oxidation reaction and improve the cyclability of the TiN anode [https://doi.org/10.1002/aenm.201502159]. In fact, there is a contribution from TiN and carbon to the capacity of the TiN/C composite anode. We have performed experiments in our lab on carbon-fiber anodes and we found that at the same condition (i.e., at the same current density of 100 mAhg-1) the capacity of carbon-fibers anode was about 200 mAhg-1 after 100 cycles, which is lower than those observed in the present work. TiN has been used as a coating material for high-capacity anodes in LIBs such as Si and SnO2 [Electrochem. Solid-State Lett., 2000, 3, 493 and New J. Chem., 2013, 37, 2096]. The TiN/Si and TiN/SnO2 composite anodes exhibited good electrochemical performance compared to the Si and SnO2 anodes. This was in part due to the good electrical conductivity, chemical and thermal stability of TiN [Electrochem. Solid-State Lett., 2000, 3, 493 and New J. Chem., 2013, 37, 2096]. We are conducting more work in our lab to explore the effect of TiN and TiN coating on the performance of LIB anodes.

This discussion was included in the revised manuscript.

We added the following paragraph to the manuscript

Figure 9 (a and b) shows the galvanostatic charge/discharge curves of H2PC(TD) and TiN(TD) anodes after 100 cycles at a current density of 100 mAg-1 within a voltage window of 0.1-3 V. Both anodes show high irreversible capacities at the first discharge (Li-insertion) cycle. The irreversible capacity may be caused by the reductive decomposition of electrolyte solution and the subsequent formation of the SEI layer at the anode surface. However, the TiN(TD) anode showed a lower irreversible capacity at the first charge cycle (44% loss in capacity) than that observed for the H2PC(TD) anode (58% loss). The capacity retention after the second cycle for both electrodes was good (almost 100 %) resulting in a high coulombic efficiency of nearly 100% indicating the formation of a stable interface (SEI layer) during subsequent charge discharge cycles.

The following Figures were added to the revised manuscript:

A new Figure 9 was added to the manuscript please see the manuscript text.

Figure 9A. Charge/discharge curves at a current density of 100 mAg-1  H2PC(TD). B. Charge/discharge curves of TiPC(TD) anode at a current density of 100 mAg-1.

Comment At least, to validate the results I would test the performances of commercial graphite in the same conditions.

The Coulombic efficiency should be reported together with the capacity values.

Response: The capacity of commercial graphite anode is 372 mAhg-1. Many results have been reported on the electrochemical performance of carbon-based anodes and their composites [Advanced Functional Materials 2006;16(18):2393-7, https://doi.org/10.1002/app.50396 and http://doi.org/10.1002/pen.24816 ]. The present work shows that the capacity of H2PC (carbon) is higher than carbon or commercial graphite anodes. This may be  due to the anode not only contains carbon but also Nitrogen on the surface, which can form defects on the anode surface after carbonization and this can result in increased capacity as it was reported in the literature [https://doi.org/10.1002/aenm.201502159; http://dx.doi.org/10.1016/j.materresbull.2016.08.040 and Mater. Futures 1 (2022) 045102]. The investigation of the effect of N2 on anode performance after the carbonization of H2PC is out the scope of the present manuscript and requires more work.  

Figure 10 was modified to include the coulombic efficiency of (TiN(TD)C and H2PC(TD) electrodes.

We added the following paragraph and Figure to the revised manuscript:

Figure 10 shows the cycle performance and Coulombic efficiency of H2PC(TD) and TiN(TD) electrodes after 100 cycles at 100 mAg–1. The H2PC(TD) anode exhibited a stable charge capacity; which is consistent with the observation that the H2PC(TD) anode exhibits a higher theoretical capacity than TiN(TD). After the 5th cycle, the capacity of the H2PC(TD) anode started to increase, and after 30 cycles, the capacity was decreased to 420 mAhg–1 and remained stable up to 100 cycles. On the other hand, the charge capacity of the TiN(TD) electrode decreased after a few cycles and then remained constant at 350 mAh g–1 after 100 cycles indicating good electrochemical stability. The TiN(TD) anode exhibits higher initial coulombic efficiency (56.1%) and better capacity retention (98.3% of the 2nd cycle) than the H2PC(TD) anode. The improvement of TiN(TD) capacity compared to commercial graphite anode or carbon fiber anodes might be due to the synergistic effect of TiN on the electrochemical performance of the carbon matrix anode. and might also be due to the good electrical conductivity and high chemical and thermal stability of TiN. However, pure TiN suffers from  irreversible oxidation reactions, which lead to  poor  cyclability [62]. For this reason, the addition of conductive carbon to TiN can prevent the oxidation reaction and improve the cyclability of TiN anode [62] In fact, there is a contribution from TiN and carbon to the capacity observed for the TiN(TD) anode. Previous experiments performed on carbon-fiber anodes under the same condition (current density of 100 mAhg-1) have shown the capacity of carbon-fibers anode was approximately 200 mAhg-1 after 100 cycles, which is lower than those observed in the present work [63-65 ]. TiN has been used as a coating material for high-capacity anodes in LIBs such as Si and SnO2 [66,67]. The TiN/Si and TiN/SnO2 composite anodes exhibited good electrochemical performance compared to the Si and SnO2 anodes. This was in part due to the good electrical conductivity, chemical and thermal stability of TiN [66,67].

We added a Figure 10 to the manuscript please see the text.

Figure 10. Electrochemical performance and coulombic efficiency of thermally carbonized metal-free phthalocyanine and titanium phthalocyanine electrode materials.

Round 2

Reviewer 2 Report

Comments and Suggestions for Authors

The Authors have carefully considered all the points raised by the Reviewer, adding data, correcting the issues and better explaining their position. Now the take-home message is clear and the manuscript is coherent. Therefore, it is suitable for publication.